



# Higher mass loss over Greenland and Antarctic ice sheets projected in CMIP6 than CMIP5 by high resolution regional downscaling EC-Earth

Fredrik Boberg[1], Ruth Mottram[1], Nicolaj Hansen[1,2], Shuting Yang[1], Peter L. Langen[3]

[1]Danish Meteorological Institute, Copenhagen Ø, DK-2100, Denmark
[2]National Space Institute, Kongens Lyngby, DK-2800, Denmark
[3]iClimate, Department of Environmental Science, Aarhus University, Roskilde, DK-4000, Denmark

*Correspondence to*: Fredrik Boberg (fbo@dmi.dk)

**Abstract.** The future rates of ice sheet melt in Greenland and Antarctica are an important factor when making estimates of the likely rate of sea level rise. Global climate models that took part in the fifth Coupled Model Intercomparison Project (CMIP5) have generally been unable to replicate observed rates of ice sheet melt. With the advent of the sixth Coupled Model Intercomparison Project (CMIP6), with a general increase in the equilibrium climate sensitivity, we here compare two versions of the global climate model EC-Earth using the regional climate model HIRHAM5 downscaling EC-Earth for Greenland and Antarctica. One version (v2) of EC-Earth is taken from CMIP5 for the high-emissions Representative Concentration Pathways (RCP8.5) scenario and the other (v3) from CMIP6 for the comparable high-emissions Shared Socioeconomic Pathways (SSP5-8.5) scenario). For Greenland, we downscale the two versions of EC-Earth for the historical period 1991-2010 and for the scenario period 2081-2100. For Antarctica, the periods are 1971-2000 and 2071-2100, respectively. For the Greenland Ice Sheet, we find that the mean change in temperature is 5.9 °C when downscaling EC-Earth v2 and 6.8 °C when downscaling EC-Earth v3. Corresponding values for Antarctica are 4.1 °C for v2 and 4.9 °C for v3. The mean change in surface mass balance at the end of the century under these high emissions scenarios is found to be -210 Gt yr$^{-1}$ (v2) and -1150 Gt yr$^{-1}$ (v3) for Greenland and +150 Gt yr$^{-1}$ (v2) and -710 Gt yr$^{-1}$ (v3) for Antarctica. These distinct differences in temperature change and particularly surface mass balance change are a result of the higher equilibrium climate sensitivity in EC-Earth v3 (4.3 K) compared with 3.3 K in EC-Earth v2 and the differences in greenhouse gas concentrations between the RCP8.5 and the SSP5-8.5 scenarios.

## 1 Introduction

The melt of ice sheets and glaciers now accounts for a greater proportion of observed sea level rise than thermal expansion (Chen et al. 2013, IPCC 2019). With around 150 million people living within 1 meter of current global mean sea level (Anthoff et al. 2006), understanding the likely rate of sea level rise is crucial for planning infrastructure and coastal development. Global climate models that took part in the fifth Coupled Model Intercomparison Project (CMIP5, Taylor et al.



2012) have generally been unable to replicate observed rates of ice sheet melt in Greenland at the present day (Delhasse et al. 2018) and estimates of sea level contributions from both large polar ice sheets are tracking the upper end of the range of estimates from these models (Slater et al. 2020). Internal variability in the Southern Ocean makes estimating Antarctic surface mass balance (SMB) complicated and can mask trends related to global warming (Mottram et al. 2020). These uncertainties in current ice sheet response from observations and models give rise to the possibility that the rate of sea level

rise over the course of the 21st century may be underestimated in current climate assessments driven by CMIP5 and earlier model intercomparisons (Slater et al. 2020; Hanna et al. 2018).

While the CMIP5 experiments were driven by the Representative Concentration Pathways (RCPs, van Vuuren et al. 2011), models in the sixth intercomparison project (CMIP6, Eyring et al. 2016) use a new set of emission and land use scenarios based on socio-economic developments, Shared Socioeconomic Pathways (SSPs, Riahi et al. 2017, O'Neill et al. 2016).

Here we use only one of the SSPs called SSP5-8.5, characterized by fossil-fueled development that is the only SSP consistent with emissions high enough to realize an anthropogenic radiative forcing of 8.5 W m$^{-2}$ in 2100. The total forcing of SSP5-8.5 at 2100 therefore matches that of the RCP8.5 used in CMIP5, but the pathway is different as is the composition in terms of different contributions. For instance, in SSP5-8.5, $CO_2$ emissions and concentrations are somewhat higher than in RCP8.5, but this is compensated for by other constituents such as $CH_4$ and $N_2O$. In this study, we compare results forced by

two versions of the EC-Earth coupled global model for RCP8.5 with EC-Earth v2 and SSP5-8.5 with EC-Earth v3. These two scenarios were chosen as they are the most similar to each other between the CMIP5 and CMIP6 experiments that have been carried out with both model versions.

Several different participating models in the latest generation of global climate models run for CMIP6 (Eyring et al. 2016) have demonstrated an increase in the equilibrium climate sensitivity (ECS) of the models compared to the previous versions

in CMIP5 (Voosen 2019, Zelinka et al. 2020). ECS is defined as the time averaged near-surface air warming in response to doubling $CO_2$ in the atmosphere relative to pre-industrial climate, after the climate system has come into equilibrium. ECS is a commonly used metric to quantify the global warming to increases in atmospheric $CO_2$ including fast feedbacks in the climate system. The higher the ECS, the greater the likelihood of the climate system reaching higher levels of global warming, the smaller the permissible carbon emissions in order to meet a particular climate target. Therefore the ECS is also

highly relevant for climate policy.

EC-Earth v3 has a higher ECS of 4.3K compared to 3.3K of EC-Earth v2 from CMIP5 due mainly to a more advanced treatment of aerosols (Wyser et al. 2020b). In this paper, we compare downscaled climate simulations from both versions for Greenland and Antarctica, run with the HIRHAM5 regional climate model to examine the impact of the higher ECS on estimates of ice sheet mass budget for both Greenland and Antarctica over the 21st century. Higher ECS leads to more rapid

atmospheric warming for a given forcing and thus enhanced rates of ice sheet melt. However, as precipitation often increases in lockstep with a warmer atmosphere, this enhanced melt may be offset to some degree by enhanced snowfall.





The SMB, sometimes also called climatic mass balance, of ice sheets and glaciers is the balance between precipitation and the melt and runoff of snow and glacier ice, accounting also for retention and refreezing within the snowpack (Lenaerts et al. 2019). SMB controls the dynamical evolution of ice sheets by driving ice sheet flow from areas of high accumulation to

regions of high ice loss. Surface melt and runoff accounts for around 50% of the ice lost from Greenland (Shepherd et al. 2019). In Antarctica, dynamical ice loss by calving and the submarine melting of ice shelves are the main sinks for ice loss and SMB processes are largely, with some exceptions, especially in the Antarctic Peninsula, mass gain processes.

As suggested by Hofer et al. (2017), SMB in Greenland, derived by dynamical downscaling of ERA-Interim reanalysis (Dee et al. 2011) with regional climate models, has a larger runoff component compared with CMIP5 models. This has been

attributed to, for instance, a cooler than observed Arctic in EC-Earth v2 by Mottram et al. (2017) or inadequate representation of Greenland blocking and the North Atlantic Oscillation (NAO) by Hanna et al. (2013). Hofer et al. (2017) and Ruan et al. (2019) also show that cloud properties in climate models are the means by which the NAO modulates ice sheet melt and inadequacies in their representation may be a further source of uncertainty within projections of ice sheet SMB in both Greenland and Antarctica.

Relatively few RCMs have been run or studied in depth for the SMB of Antarctica and results used in international ice sheet modelling intercomparisons have by and large focused on using results from the MAR and RACMO  (e.g. Lenaerts et al. 2016, Agosta et al., 2013, 2019; Kittel et al., 2018; Van Wessem et al., 2015, 2018; ). Results of a recent intercomparison of regional models all forced by ERA-Interim (Mottram et al. 2020) show a wide spread of estimates of present day SMB (from 1960 to 2520 Gt yr$^{-1}$) related to in large part to different resolutions and precipitation schemes. However, a comparison of

future projections from studies by Ligtenberg et al. (2013) and Hansen (2019) suggests that on the scale of decades to centuries a clear upward trend in SMB with large interannual and decadal variability is expected due to enhanced snowfall in a warmer climate.

Both the Greenland and the Antarctic ice sheets are important to understand in estimating sea level rise due both to their absolute possible contribution to sea level and for the different timescales and processes that could drive their disintegration.

The Antarctic Ice Sheet stores approximately 90% of Earth's freshwater, a potential contribution to mean sea level of 58 m (Fretwell et al. 2013). Thus, the Antarctic Ice Sheet has the potential to be the single largest contributor to future sea level rise. The Greenland Ice Sheet contains around 7 m of mean sea level rise (Aschwanden et al. 2019) and has in the last two decades seen increasing mass loss (450–500 Gt yr$^{-1}$) due to both extreme surface melt events and enhanced calving from outlet glaciers (Mankoff et al. 2019).

Recent projections from both Greenland and Antarctica have started to include coupled climate and dynamical ice sheet models from both intermediate complexity models as well as fully coupled regional and global models (Robinson et al. 2012; Vizcaino et al. 2013;  Levermann et al. 2020; Le Clec'h et al. 2019; Sloth Madsen et al. in prep). However, most





studies still rely on offline ice sheet models forced by higher resolution regional climate models that downscale from global models. In Antarctica, as most ice loss is dynamically driven, SMB is primarily used to provide accurate forcing for ice sheet
models. Ice Sheet Model Intercomparison Project for CMIP6 (ISMIP6) models (Goelzer et al. 2018) suggest a wide spread in projections of sea level rise for Greenland from 70 to 130 mm (Goelzer et al. 2020) including both dynamical and SMB contributions calculated from several different GCMs.

In this paper we investigate the differences between two different versions of the global climate model EC-Earth, using an identical version of the regional climate model HIRHAM5, for the Greenland and Antarctica ice sheets (see Figure 1). The
two EC-Earth models are EC-Earth v2.3 and EC-Earth v3.3 (hereafter referred to as EC-Earth2 and EC-Earth3) and are run for CMIP5 and CMIP6, respectively. The comparison focuses on temporal changes (end of century relative to a reference period) in temperature, precipitation and the surface mass balance.

In Section 2 we introduce the model domains, the two versions of the global climate model EC-Earth as well as the regional climate model HIRHAM5. In Section 3 we present, using time slice experiments and for both Greenland and Antarctica,
changes in temperature and precipitation using the two versions of EC-Earth, followed by the resulting changes in surface mass balance for both ice sheets. The paper ends with a discussion in Section 4 and a conclusion in Section 5.

## 2 Methods

Here we compare regionally downscaled climate simulations for Greenland and Antarctica (see Figure 1) run with two different versions of EC-Earth and an identical version of the HIRHAM5 RCM. The two EC-Earth models, i.e, EC-Earth
v2.3 and EC-Earth v3.3 (hereafter referred to as EC-Earth2 and EC-Earth3) are run for CMIP5 and CMIP6, respectively. For reasons of computational cost we run four time slice experiments with HIRHAM5 driven with EC-Earth forcings for each domain. For Greenland, these cover the period 1990-2010 with historical forcing with both versions of EC-Earth and the period 2080-2100 with CMIP5 RCP8.5 for EC-Earth2 and CMIP6 SSP5-8.5 with EC-Earth3. The historical forcing ends in 2005 for CMIP5 and therefore for the last 5 years of the 1990-2010 period we use RCP4.5 scenario forcing.


For Antarctica, the time slice experiments cover the period 1970-2000 with historical forcing and the period 2070-2100 with RCP8.5 and SSP5-8.5. The first year in each time slice experiment is used for spin-up of atmospheric conditions and not included in the analysis. Also, for the four time slice experiments in Greenland we include an offline spin-up routine of subsurface conditions running for 100 years (Langen et al. 2017). The output from HIRHAM5 is used to calculate the SMB
of the ice sheets over these periods in order to be able to compare the different forcings. We use the HIRHAM5 downscaling to give a better representation of the surface energy balance over the ice sheet with all radiative and turbulent heat fluxes represented as well as surface snow properties that allow the retention and refreezing of meltwater.



EC-Earth is a global climate model evolving from the seasonal forecast system of the ECMWF (Hazeleger et al., 2010) and
developed by a large European consortium. EC-Earth2 is the model used to contribute to CMIP5 and is based on the
ECMWF integrated forecasting system (IFS) cy31r1, the NEMO version 2 ocean model and the sea ice model LIM2
(Hazeleger et al, 2012). EC-Earth2 is run on a spectral resolution of T159 (equivalent to ~125 km) and 62 vertical levels up
to 5 hPa for the atmosphere, and a 1° x 1° tripolar grid with 46 vertical levels for the ocean and sea ice. The new generation
of the EC-Earth model is a full Earth System model and has been developed to perform CMIP6 experiments. A detailed
description of this model is under preparation (Döscher et al., 2020). However, the CMIP6 historical and SSP5-8.5
experiments used in the downscaling in this study were performed with only the GCM configuration i.e, EC-Earth3. EC-
Earth3 has upgraded all components of EC-Earth2, with the IFS cy36r4 for the atmosphere model, the NEMO version 3.6 for
the ocean with the sea ice model LIM3 embedded. All these component models have improved the representation of the
physical processes greatly. The EC-Earth3 also runs at a higher resolution than the EC-Earth2. The spatial resolution of the
atmosphere is about 80 km horizontally (T255) and 91 vertical levels up to 0.01 hPa for the atmosphere. The ocean model
uses the same 1° x 1° tripolar grid as the EC-Earth2 but with 75 vertical levels. The EC-Earth contributed to CMIP5 and
CMIP6 historical and scenario experiments with ensembles of 15 and 25 members in total, performed on various platforms
by respective consortium members. The differences among these members are only on the initial states which are taken from
different snapshots in a 500 year long control run under the pre-industrial condition (Taylor et al, 2012; Eyring et al, 2016).
The simulations used in this study were the members r3i1p1 for the CMIP5 and r5i1p1f1 for the CMIP6, carried out at the
Danish Meteorological Institute. Figures 2a and 2b show the 1991-2010 mean temperature relative to ERA-Interim for EC-
Earth2 and EC-Earth3, respectively. The cold bias over Greenland for EC-Earth2 in Figure 2a is not present for EC-Earth3 in
Figure2b. EC-Earth3 has, however a warm bias over Antarctica. Figures 2c and 2d show the difference in the change in 2m
temperature and sea surface temperature, respectively, between the EC-Earth3 using SSP5-8.5 and the EC-Earth2 using
RCP8.5 at the end of the century relative to the reference period. For 2m temperature in Figure 2c we see a positive
difference for both Greenland and Antarctica: between 1 and 3°C along the coastal regions for Greenland and about 1°C in
the central parts of Antarctica. There is also a clear difference in sea surface temperature change between the two versions of
EC-Earth in Figure 2d: between 1 and 3°C along the coast of Greenland and between 1 and 2°C along the coast of
Antarctica. These differences in both atmospheric temperature and sea surface temperature are reflected in differences in
end-of-winter sea ice extent shown in Figure 3.

The HIRHAM5 regional climate model (Christensen et al. 2006) is based on the HIRLAM7 weather forecasting model
(Undén et al. 2002) where the physical routines have been replaced by those within the ECHAM5 climate model (Roeckner
et al. 2003). HIRHAM5 uses 31 atmospheric levels and for the Greenland domain, the model is run at a resolution of 0.05°
(about 5.5 km) with 20 year long time slices while the Antarctica simulation is run at a resolution of 0.11° (about 12.5 km)
with 30 year long time slices. The HIRHAM5 model has previously been validated against observations for Greenland (e.g.
Boberg et al. 2018; Langen et al. 2017) and Antarctica (Mottram et al. 2020, Hansen 2019). Boberg et al. (2018) showed that





monthly means of observed temperature on the Greenland ice sheet compare well with the EC-Earth2 downscaling using
HIRHAM5 for the period 1993-2010 with a mean bias between +1 and -2°C. Langen et al. (2017) compared 1041 SMB
observations from 351 locations in the ablation area of the Greenland Ice Sheet with an ERA-Interim driven HIRHAM5
simulation and found a regression slope of 0.95, a correlation coefficients of 0.75 and a mean bias of -3%, indicating only
slightly underestimated net mass loss rate. Moreover, comparing to 68 ice cores in the accumulation area of the Greenland
Ice Sheet, they found the simulated mean annual accumulation rate to have a -5% bias and a correlation coefficient of 0.9.
Mottram et al. (2020) showed, using station observations, that ERA-Interim forced HIRHAM5 simulations have a cold bias
of 1−2°C for Antarctica.

## 3 Results

### 3.1 Modelled Temperature

Figures 4a and 4c show the annual mean change in 2m temperature for Greenland and Antarctica respectively using
HIRHAM5 downscaled with EC-Earth3 for 2081-2100 and 2071-2100 for the SSP5-8.5 scenario relative to the 1991-2010
and 1971-2000 historical runs. Figures 4b and 4d show the difference between the changes given in Figures 4a and 4c and
the equivalent change using EC-Earth2 for the same time periods but using the RCP8.5 forcing scenario. Therefore positive
values in Figures 4b and 4d do not imply that the scenario period in the EC-Earth3 SSP5-8.5 downscaling is warmer than the
scenario period in the EC-Earth2 RCP8.5 downscaling - just that the *change* in temperature is larger from the historical
period to the SSP5-8.5 runs compared with the change between the historical simulation and the RCP8.5 runs. The mean
change in temperature over the ice sheet is 5.9 °C for Greenland using EC-Earth2 and 6.8 °C using EC-Earth3. For
Antarctica the values are 4.1 °C using EC-Earth2 and 4.9 °C using EC-Earth3.

For Greenland (Figure 4b), the change in temperature for the EC-Earth3 run using the SSP5-8.5 scenario is shown to be
higher for most of the domain compared with the change in temperature for the EC-Earth2 run using the RCP8.5 scenario.
For Antarctica (Figure 4d), we see similar values as for Greenland except for the eastern part of Antarctica and the western
side of the peninsula. This pattern is probably related to the temperature change difference in the GCMs seen in Figure 2c
along part of the coastal stretches of Antarctica which in turn could be explained by a change in model bias and/or as a result
of aerosol differences between the two GCM versions. As the phase of the southern annular mode (SAM) and also controls
the spatial variability in precipitation and temperature on annual to decadal scales in Antarctica , the pattern may also reflect
different phases of the SAM in the two versions that are, at least in part a result of internal variability rather than climate
forcing (Fogt and Marshall, 2020).



### 3.2 Modelled Precipitation

For precipitation, we see a positive relative change for both domains (Figure 5a and 5c) using EC-Earth3 and the SSP5-8.5 scenario when downscaling using HIRHAM5. When comparing the difference in relative changes in precipitation (Figure 5b and 5d) we see negative values for the eastern part of the domains and positive values for the western parts. These east-west

patterns are reminiscent of those in the differences in temperature changes shown in Figure 4b and 4d and in turn are similar to spatial patterns shown in ice core records by Medley and Thomas (2019) which they relate to SAM variability. This suggests that understanding internal variability in global models is important for interpreting SMB projections in Antarctica.

### 3.3 Modelled SMB

Figure 6 shows the change in SMB for Greenland (panels a and b) and Antarctica (panels c and d). Figure 6a and 6c shows

downscaled EC-Earth2 for the RCP8.5 scenario while Figure 6b and 6d shows downscaled EC-Earth3 for the SSP5-8.5 scenario, all relative to the historical periods (see Table 1). For EC-Earth2 we get a change (2081-2100 relative to 1991-2010) in SMB of -210 Gt yr$^{-1}$ for the entire Greenland Ice Sheet with areas along the western part displaying changes in the range -2 to -1 m yr$^{-1}$ using the simplified equation for surface mass balance: SMB = precipitation - evaporation - sublimation - surface runoff.  This change in SMB of -210 Gt yr$^{-1}$ obtained using HIRHAM5 forced with EC-Earth2 is identical to the

cumulative value of -210 Gt yr$^{-1}$ obtained using a more realistic approach (not shown) with an updated version of the subsurface with more detailed physics using many more layers (Langen et al. 2017).

For EC-Earth3 (Figure 6b) almost the entire Greenland ice sheet shows a negative change (2081-2100 relative to 1991-2010) in the SMB with values well below -2 m yr$^{-1}$ along the margin. Over the thirty year period at the end of the century for which the model is run, the accumulated SMB anomaly is -1150 Gt yr$^{-1}$ equivalent to an additional 3.2 mm of sea level rise

per year from the Greenland ice sheet at the end of the century, in line with estimates published by Hanna et al. (2020). We also note that the area in the southeast part of the Greenland Ice Sheet with positive contributions for the EC-Earth2 run in Figure 6a is no longer present for the EC-Earth3 run in Figure 6b. For Antarctica, we get a change (2071-2100 relative to 1971-2000) in SMB of 150 Gt yr$^{-1}$ for the EC-Earth2 simulation (Figure 6c) and a value of -710 Gt yr$^{-1}$ for the EC-Earth3 simulation (Figure 6d). This is equivalent to a sea level rise contribution of around 2.0 mm per year from SMB processes,

assuming that the melt originates from the grounded ice sheet only. Importantly, the location of the negative SMB in the model coincides with the vulnerable west Antarctic outlet glaciers whose destabilisation could lead to rapid retreat and dynamical ice loss, multiplying many times the effects of the enhanced ice sheet loss.

When looking at yearly sums of the two ice sheet components, precipitation-minus-sublimation and runoff, we can further

study the differences between EC-Earth3 and EC-Earth2 (v3 minus v2) for our two model domains (cf. Table 2). For Greenland during the historical period 1991-2010 (Figure 7a), the runoff component for the EC-Earth3 downscaled simulation is about 230 Gt yr$^{-1}$ larger than for EC-Earth2 while the precipitation minus sublimation component has a mean

difference of about 120 Gt yr$^{-1}$ with relatively large variations for both simulations. For Greenland during the scenario period 2081-2100 (Figure 7b), the two simulations show a similar difference (now a mean difference of 105 Gt yr$^{-1}$) with respect to the historical period 1991-2010 for the precipitation minus sublimation component, whereas runoff shows a steady increase in the difference between the simulations reaching in excess of 1100 Gt yr$^{-1}$ at the end of the century.

For Antarctica during the historical period 1971-2000 (Figure 7c), we see a mean difference of about 560 Gt yr$^{-1}$ for precipitation minus sublimation (cf. Table 2) between the two simulations but for the runoff component, the difference is about 400 Gt yr$^{-1}$ and only small variations are seen, especially for the EC-Earth2 run. For Antarctica during the scenario period 2071-2100 (Figure 7d), we see that the gap between both precipitation minus sublimation as well as runoff increases with time reaching a difference of more than 1200 Gt yr$^{-1}$ and 2600 Gt yr$^{-1}$ by the end of the century, respectively.

As the large differences between model versions in ΔSMB (940 Gt for Greenland and 860 Gt for Antarctica) are dominated by differences in runoff changes rather than precipitation changes (see Table 2), we attribute them to the approximately 1 °C higher end-of-century warming in both Greenland and Antarctica for EC-Earth3 relative to EC-Earth2.

## 4 Discussion

Our results show that for two different versions of the driving global model, substantial differences arise in ice sheet surface mass balance at the end of the century when driven by similar greenhouse gas emissions pathways. The runoff and precipitation rates at the end of the century over both Greenland and Antarctica are higher, and likely enhanced by, the higher temperatures projected under SSP5-8.5 than RCP8.5. The higher temperatures in the EC-Earth3 driven downscalings for the SSP5-8.5 scenario compared with those for the EC-Earth2 driven downscalings for the RCP8.5 scenario are partly caused by a higher equilibrium climate sensitivity (4.3 K compared with 3.3 K in EC-Earth2). The difference between the greenhouse gas emission pathways in SSP5-8.5 and RCP8.5 do also play an important role, however. Gidden et al. (2019) found that the radiative forcing in SSP5-8.5 matched that of RCP8.5 closely but that there were clear differences between the individual greenhouse gas components of the forcing as well as the aerosols. Wyser et al. (2020a) compared an EC-Earth run in CMIP6 (called EC-Earth3 Veg) and the CMIP5 EC-Earth run and concluded that 50% or more of the end of century global temperature increase going from CMIP5 to CMIP6 was due to changes in the greenhouse gas concentrations rather than model changes

The clear downward trend in SMB found when downscaling EC-Earth3 for Antarctica is at odds with previous studies. Ligtenberg et al. (2013) downscaled ECHAM5 using the A1B scenario while Hansen (2019) downscaled EC-Earth2 using the RCP8.5 scenario, both showing upward trends in SMB for Antarctica due to enhanced snowfall in a warmer climate. In

EC-Earth3 the increase in melt due to the markedly warmer climate appears to outpace the increase in precipitation,
suggesting that there is greater uncertainty in the future SMB of Antarctica than previously identified.

For this study, only one RCM has been used when comparing the downscaling of two GCMs. The results presented in this study would therefore benefit from future expansion to a multi-model and multi-member ensemble. However, the HIRHAM5 model has been used for downscaling EC-Earth for both Greenland and Antarctica in a number of studies
(Langen et al. 2017, Boberg et al. 2018, Hansen 2019, Mottram et al. 2020) and the model output has been validated thoroughly giving it validity for climate modelling as a single member for polar conditions.

Our results for Greenland are in line with previous work by Van Wessem et al. (2018) and Agosta et al. (2019). These models also showed a general increase in melt and runoff rates for Greenland and Antarctica when driven by selected
CMIP6 models compared with CMIP5. As Kittel et al. (2020) also show, in our results the increasing melt is particularly apparent on the ice shelves during this century rather than the grounded ice sheet, which mitigates the contribution to sea level rise to some extent. Our conclusions in Antarctica are also in line with results from Holland et al. (2019), who showed that the more extreme the climate scenario in CMIP5, the greater the warming in West Antarctica due to persistent mean westerly shelf-break winds.


Bracegirdle et al. (2015) used 37 CMIP5 models and showed, due to a large intermodel spread in sea ice area that the change in temperature using the RCP8.5 scenario for Antarctica was in the range 0 to 6 ℃ while the change in precipitation was in the range 0 to almost 40%. This large model spread for future climate change for Antarctica clearly shows the importance of using large model ensembles for climate projections. Analysis of the CMIP6 ensemble for Antarctic sea ice by Roach et al.
(2020) showed some improvement in regional sea ice distribution and historical sea ice extent as well as a slight narrowing of the multimodel ensemble spread in CMIP6 compared to CMIP5. Although the wide spread in projections indicates that a large multi-model ensemble is desirable, comparing two slightly different versions of the same model is helpful to determine which changes may be affected by the difference in the driving models as well as the emissions pathways. The importance of sea surface temperature and sea ice extent to SMB in Antarctica, especially in coastal regions (Kittel et al. 2018) means that
variability in ocean and sea ice representation in model projections has large implications for SMB estimates.

## 5 Conclusion

Due to a higher ECS in the driving GCM EC-Earth3 within CMIP6 compared with the driving GCM EC-Earth2 within CMIP5 together with changes in greenhouse gas concentrations between the RCP8.5 and the SSP5-8.5 scenarios, we find larger changes in both temperature and precipitation for both model domains in the end-of-century scenario runs compared
with the historical simulations. These differences lead to important changes over the polar ice sheets with a change in SMB

of around -1150 Gt yr$^{-1}$ for Greenland and -710 Gt yr$^{-1}$ for Antarctica at the end of the century. Comparing these numbers with the ones obtained from the older EC-Earth2 runs (-210 Gt yr$^{-1}$ for Greenland and +150 Gt yr$^{-1}$ for Antarctica), suggests that for very high emissions pathways, considerable uncertainty still exists for sea level rise contributions from the polar ice sheets due to climate change. The difference between these two versions corresponds to a sea level rise difference of 2.6 mm

per year from Greenland and 2.3 mm per year for Antarctica at the end of the century compared with earlier estimates based on EC-Earth2. It is therefore very important to understand and accurately estimate any changes in the surface mass budget of both the Greenland and the Antarctic ice sheets for the 21st century when estimating the impact of sea level rise on coastal infrastructure and development.

We find that it is difficult to directly compare the downscalings of EC-Earth2 and EC-Earth3 since the forcing conditions are not equal due to revised greenhouse gas concentration scenarios, however this allows us to demonstrate the potentially wide uncertainties on SMB estimates. Moreover the role of natural variability and the impact of climate change on regional circulation patterns that affect SMB are clearly areas that need more research in the future. The results presented here using EC-Earth3 within CMIP6 are therefore important to consider when communicating to the adaptation and mitigation

communities.

**Author contribution**

FB, RM, NH and PLL designed the experiments and FB carried them out. FB performed the HIRHAM5 simulations. SY developed the model code for EC-Earth and performed the simulations. FB prepared the manuscript with contributions from all co-authors.

**Acknowledgements**

This work is supported by the NordForsk-funded Nordic Centre of Excellence project (award 76654) *Arctic Climate Predictions: Pathways to Resilient, Sustainable Societies (ARCPATH)*. This work has also been supported by the Horizon 2020 EUCP EUropean Climate Prediction system under Grant agreement no. 776613. The work is also supported by the project "Producing RegIoNal ClImate Projections Leading to European Services" (PRINCIPLES, C3S_34b Lot2), part of the

Copernicus Climate Change Service (C3S) provided by the European Union's Copernicus Programme and managed by the European Commission. This publication was supported by PROTECT. This project has received funding from the European Union's Horizon 2020 research and innovation programme under grant agreement No 869304, PROTECT contribution number XX. The authors would also like to acknowledge the support of the Danish state through the National Centre for Climate Research (NCKF).




## Data Availability

HIRHAM5 simulation data for all 8 time slice simulations are available upon request to the author. EC-Earth2 and EC-Earth3 data are available on the Earth System Grid Federation portals (eg. https://esg-dn1.nsc.liu.se/projects/esgf-liu/).

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

.





| Domain | Resolution | EC-Earth Forcing | Period |
|---|---|---|---|
| Greenland | 0.05° | v2 historical | 1990-2010 |
| | | v2 RCP8.5 | 2080-2100 |
| | | v3 historical | 1990-2010 |
| | | v3 SSP5-8.5 | 2080-2100 |
| Antarctica | 0.11° | v2 historical | 1970-2000 |
| | | v2 RCP8.5 | 2070-2100 |
| | | v3 historical | 1970-2000 |
| | | v3 SSP5-8.5 | 2070-2100 |

**Table 1: List of all 8 time slice experiments. The Greenland runs are 20 years long while the runs for Antarctica are 30 years long,**
**not counting the first spin-up year in each experiment.**





| Domain | GCM | Period | Precip | Subl | Runoff | SMB | ΔSMB | $\delta$(ΔSMB) |
|--------|-----|--------|--------|------|--------|-----|------|----------------|
| Greenland | EC-Earth2 | 1991-2010 | 728 | 26 | 170 | 532 | -210 | -936 |
| | | 2081-2100 | 1045 | 32 | 691 | 322 | | |
| | EC-Earth3 | 1991-2010 | 850 | 24 | 397 | 429 | -1146 | |
| | | 2081-2100 | 1125 | 7 | 1836 | -717 | | |
| Antarctica | EC-Earth2 | 1971-2000 | 3331 | 220 | 321 | 2790 | 148 | -861 |
| | | 2071-2100 | 4276 | 280 | 1058 | 2938 | | |
| | EC-Earth3 | 1971-2000 | 3947 | 271 | 732 | 2943 | -714 | |
| | | 2071-2100 | 5166 | 319 | 2618 | 2230 | | |

**Table 2. SMB components precipitation, sublimation and surface runoff for all 8 time slice experiments in Gt yr$^{-1}$. ΔSMB is the temporal change between the scenario period and the reference period. $\delta$(ΔSMB) is the model difference in ΔSMB between EC-Earth3 and EC-Earth2.**



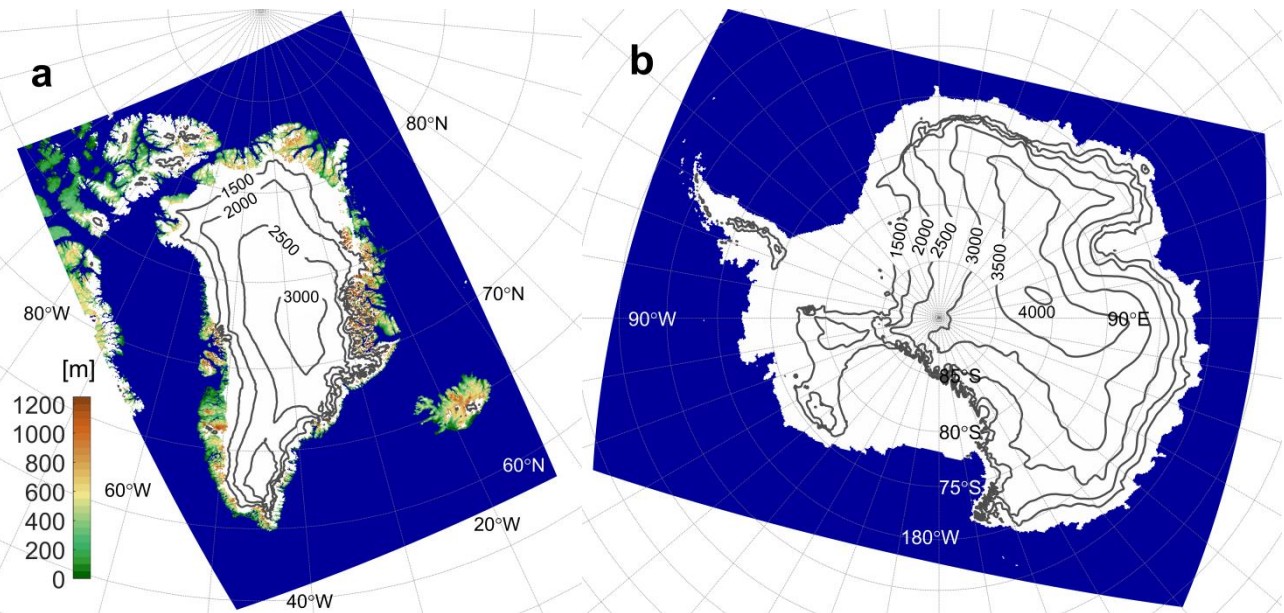

**Figure 1. Topography for the two model domains. Sea points are given in blue, non-glacial land grid points are given in green and brown while glacial points are given in white with surface elevation contour lines added. The Greenland domain (a) has a model resolution of about 5.5 km (0.05°) while the Antarctica domain (b) has a model resolution of about 12.5 km (0.11°). Note that the Antarctic ice shelves are counted as land in the analysis and included when making ice sheet means of temperature, precipitation and surface mass balance.**





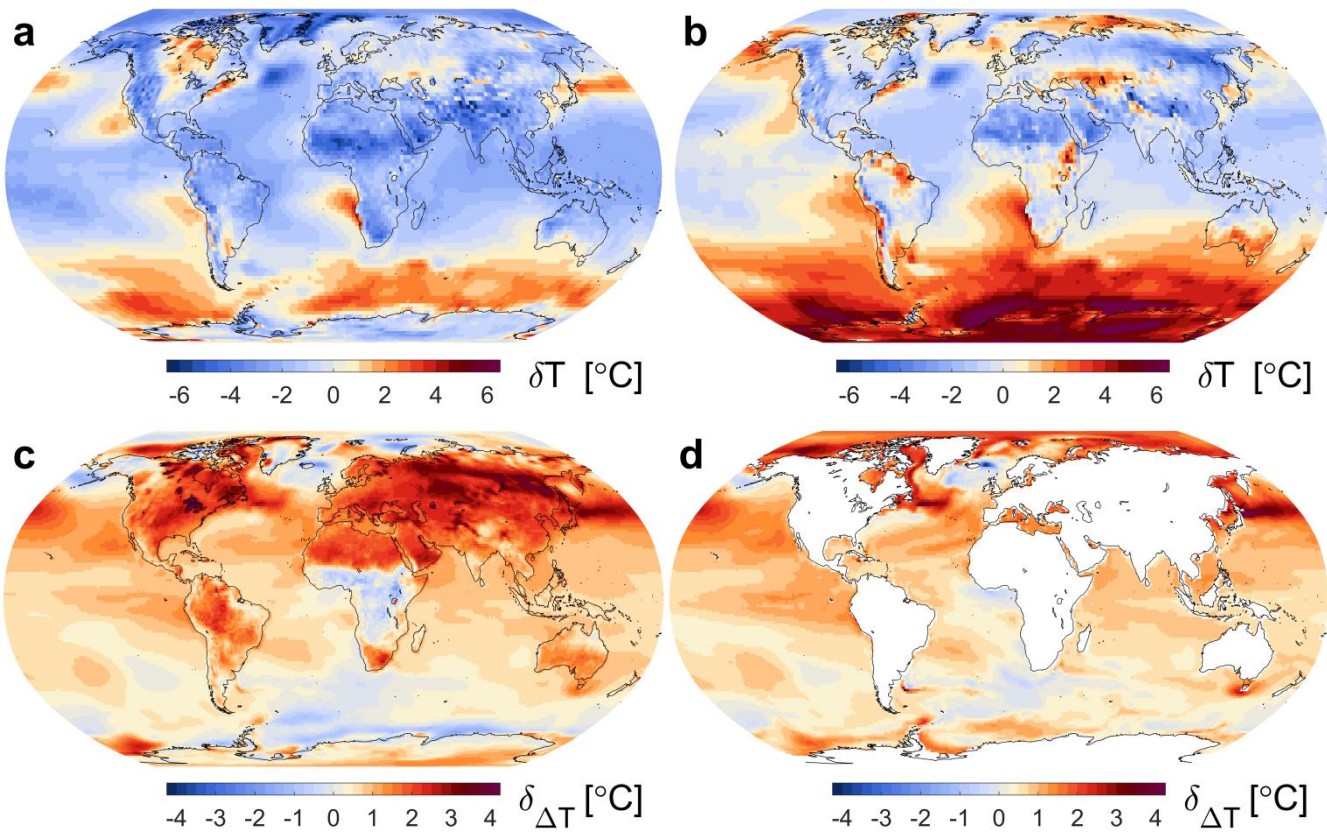

**Figure 2.** Temperature bias relative to ERA interim for 1991 to 2010 for EC-Earth2 (a) and EC-Earth3 (b). Difference in the change in 2m temperature (c) and sea surface temperature (d) for EC-Earth3 using SSP5-8.5 relative to EC-Earth2 using RCP8.5 for the 2081-2100 period relative to the 1991-2010 historical period.




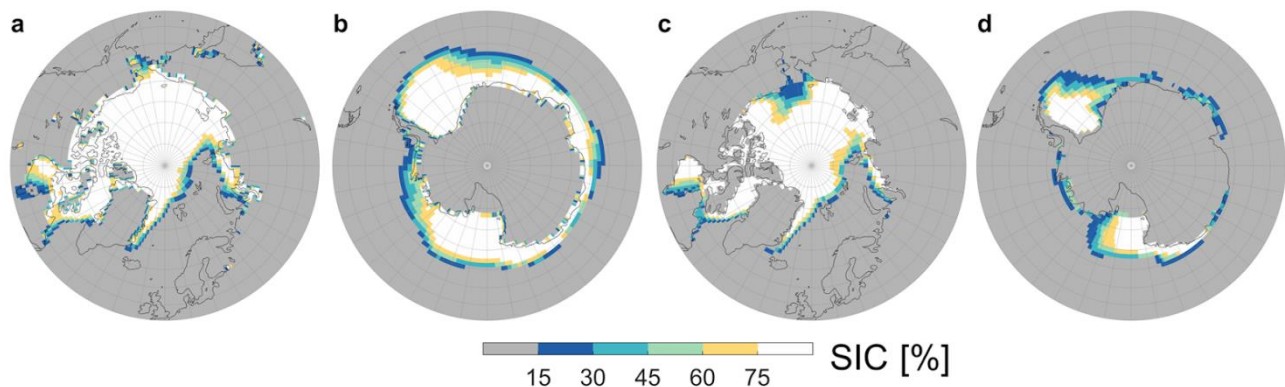

**Figure 3. Mean end-of-winter sea ice extent for the 2081-2100 period. Panels a and b are for the RCP8.5 scenario using Ec-Earth2 and panels c and d are for the SSP5-8.5 scenario using EC-Earth3. Panels a and c are for the month of March while panels b and d are for the month of September.**






**Figure 4. Change in 2 m temperature for Greenland for 2081-2100 relative to 1991-2010 for the EC-Earth v3 SSP5-8.5 scenario (a). Difference in the change in 2 m temperature for EC-Earth3 SSP5-8.5 relative to EC-Earth2 RCP8.5 (b). Change in 2 m temperature for Antarctica for 2071-2100 relative to 1971-2000 for the SSP5-8.5 scenario (c). Difference in the change in 2 m temperature for SSP5-8.5 relative to RCP8.5 (d). Note that the colorbar limits in panels a and c differ.**






**Figure 5. Relative change in total precipitation for Greenland for 2081-2100 relative to 1991-2010 for the EC-Earth3 SSP5-8.5 scenario downscaling (a). Difference in the relative change in total precipitation for Greenland for the EC-Earth3 SSP5-8.5 relative to EC-Earth2 RCP8.5 downscaling (b). Relative change in total precipitation for Antarctica for 2071-2100 relative to 1971-2000 for the EC-Earth3 SSP5-8.5 scenario downscaling (c). Difference in the relative change in total precipitation for Antarctica for the EC-Earth3 SSP5-8.5 relative to EC-Earth2 RCP8.5 downscaling (d). Note the differences in colorbar limits.**




**Figure 6. Change in surface mass balance for Greenland for the period 2081-2100 relative to 1991 to 2010 for the EC-Earth2 driven run using RCP8.5 (a) and the EC-Earth3 driven run using SSP5-8.5 (b). Change in surface mass balance for Antarctica for the period 2071-2100 relative to 1971 to 2000 for the EC-Earth2 driven run using RCP8.5 (c) and the EC-Earth3 driven run using SSP5-8.5 (d). Units are meter water equivalent per year. Green color represents non-glacial land grid points (cf. Figure 1).**





**Figure 7.** Integrated values of precipitation minus sublimation (in red using the left y-axis) and surface runoff (in blue using the right y-axis) for Greenland (panels a and b) and Antarctica (panels c and d) of HIRHAM5 downscalings of EC-Earth. EC-Earth2 is marked with diamonds and EC-Earth3 is marked with circles.