# Peer review of "Higher mass loss over Greenland and Antarctic ice sheets projected in CMIP6 than CMIP5 by high resolution regional downscaling EC-Earth"

_The Cryosphere, 2020_

## Referee Comment (RC1) · Anonymous Referee #1 · 5 Jan 2021

The paper basically downscales one member of the GCM EC-Earth-version 2 and one member of the GCM EC-Earth-version 3 for two different timeslots for Greenland and Antarctica in order to study how temperature and surface mass balance might change in the future in a high-end warming scenario. Although this is an interesting exercise and the question of how the mass loss differs in CMIP6 compared to CMIP5 is certainly relevant, I have several major objections that need to be resolved, before I can recommend the paper for publication.

First of all it is argued that there is a higher mass loss for both ice sheets in CMIP6 compared to CMIP5. However, such a bold statement cannot be made by downscaling

only one member of each ensemble. It is therefore necessary to do an analysis of a good set of CMIP5 and CMIP6 members, using different GCMs. Is the increase in the Antarctic and Greenland temperature increase (and surface mass loss) in CMIP6 compared to CMIP5 a general feature of these GCMs or is it more pronounced in EC-Earth compared to the other members. By not comparing to the other GCMs, wrong conclusions might be made about how CMIP6 differs from CMIP5.

Secondly, the authors find a large increase in warming and surface mass loss in CMIP6 than CMIP5. However, decadal variability of precipitation on the ice sheets is large and comparing a 20-year and 30-year time slots might therefore still be affected by natural variability. The authors use the DMI member of EC-Earth and it is nowhere discussed how this member relates to the other EC-Earth members. Such an analysis is necessary to better understand the contribution of natural variability compared to antropogenic forcings and climate sensitivity that might be different in CMIP5 compared to CMIP6.

As far as I understand, only the control simulation of the HIRHAM downscaled EC-Earth-version 2 has been performed (in previous papers). A basic evaluation of the HIRHAM downscaled EC-Earth-version 3 would be needed to better understand the model biases, especially given the strong warm bias in Antarctica for the present-day in the GCM.

If I understand correctly, the surface mass balance is taken over the entire ice sheet and not only the grounded ice sheet. If so, this should be changed – only the surface mass loss (and gain) over the grounded ice is relevant (as the authors are aware of).

Minor comments:

Since there is no dynamic ice sheet model, I suggest to consistently talk about surface mass loss instead of mass loss (see title for example).

For Antarctica only 1 year of spin-up is used. This is probably not enough for the

regions with substantial mass loss. Can you comment on this?

P5l133 "All these component models have improved the representation of the physical processes greatly." This statement seems optimistic given the large deterioration in EC-Earth-v3 for Antarctica. Apart from that the statement is too unspecific.

A suggestion would be to put more focus on the differences between the temperature and precipitation change in the regional model compared to the GCM. For the run-off this might not be possible, but this is – to some extent – driven by the two previous variables. An interesting question to discuss in some more detail would be how the larger temperature and precipitation change in CMIP6 translates to surface mass loss.

---

## Referee Comment (RC2) · Anonymous Referee #2 · 21 Jan 2021

**General comments**

The authors propose an assessment of the surface mass balance of the Greenland and Antarctic ice sheets at the end of the century through the downscaling of the CMIP5 and CMIP6 versions of EC-EARTH under a high-emission scenario with the regional climate model HIRHAM5. Although the paper is nicely written and fits within the current framework of reducing the uncertainties in the contribution of ice sheets to sea level rise due to the large variety of models and trajectories of greenhouse gas emissions, I think however that the paper is still at a too early stage of development and contains too many inaccuracies to correct and issues to fix to consider publication.

Although the downscaling of several GCMs requests a large amount of resources, one weakness of the paper is that only one member of one GCM from each CMIP5 and CMPI6 ensemble is used here. In that sense, this work addresses one very thin part of a much wider spectrum of future possibilities and is not particularly innovative in regard to recently published works on the same topic but based on several downscalings of GCMs (Ligtenberg et al., 2013; Fettweis et al. 2013; Kittel et al., 2020; Hofer et al., 2020), or several members (Noël al., 2021). For that reason I agree with RC1 that the title is not particularly well chosen. But the strongest weakness of the paper is the lacking, though essential, evaluation of the downscaled products over the current period. In particular, the extremely high runoff rates (> 700 Gt over 1970-2000) in Antarctica compared to preexisting model estimations (< 5 Gt), as well as high compensating snowfall rates, are not discussed at all and could be linked to the respective performances of EC-EARTH and HIRHAM5. In this context, the paper in its current form essentially consists in a comparison of present and future SMB estimates differing sometimes very significantly from the literature, with no evaluation, and whose plausibility cannot be assessed. This is yet essential. The paper requires a thorough evaluation and discussion of plausibility for a better understanding of the model results, particularly in terms of surface processes (another aspect entirely missing), especially before using them to support, for instance, that *there is greater uncertainty in the future SMB of Antarctica than previously identified*.

Too often in the paper the argumentation contains shortcuts, is in some places incomplete or not supported by the references called, requires demonstration form additional analysis and correction for inaccuracies, and the methodology could be generally much more thorough. Before potential resubmission, it is necessary that the authors carefully revise their manuscript according to each of these serious issues, for which I tried to provide a non-exhaustive list below. I also strongly encourage all the authors to carefully read the final version of the paper.

**Major/minor comments**

- P1L29: See Fettweis et al. (2013; See their Fig. 4b) instead of Delhasse et al. (2018) for an appropriate reference.

- P2L32: The way Mottram et al. (2020) could be used to justify this statement is not clear to me. Could you expand on it? Do you mean natural climate variability in the southern hemisphere? Estimates from model and/or observations ? Internal variability is not the only reason, model physics also, as well as sparsity, discontinuity and uncertainties of both ground-based and satellite observations.

- P2L55: Note that an ECS of 4.3 K is out of the 66% range for the Bayesian calculation of Sherwood et al. (2020) and is close to the upper bound of 4.5 K under plausible robustness tests, that is in the upper range of likely ECSs (if not unlikely, see Zhu et al., (2020)). This may help in discussing the likelihood of EC-Earth v3 projections, regarding its ability to represent the current climate over high latitudes and the plausibility of the associated strong (downscaled) contribution of runoff, especially for Antarctica, which is missing and yet indispensable to such an exercise.

- P2L59: Correct for ice sheet surface mass budget.

- P3L62: SMB is an important notion of the paper and needs to be properly defined. Runoff is by definition melt accounting also for retention and refreezing within the snowpack, so accounting for all this terms together in the definition of SMB is redundant. Surface sublimation and erosion/deposition of snow through drifting-snow processes, which are likely the main current ablation components at the surface of the Antarctic ice sheet (e.g., Mottram et al., 2020), are also missing.

- P3L67: "SMB processes are largely [...] mass gains processes". This phrasing is confusing. Precipitation is the dominant term and the only positive term of the SMB (with minor contributions from frost and drifting-snow deposition): not "are", but "lead to mass gain". SMB is the resultant of processes of mass gain and loss, and the balance between all of them is mostly positive.

- P3L80: See also Agosta et al. (2013) and Kittel et al. (2020).

- P4Section 2: Could you justify the two (different) historical periods for both ice sheets?

- P4L122 : Add surface sublimation. Missing drifting-snow processes should also be mentioned here. The phrasing is a bit clumsy. Surface snow properties lead to but does not allow retention and refreezing within the snowpack.

- P6 Section 3: Evaluation of downscaled EC-EARTHv2 and 3 products is inevitably required for the current climate.

- P3L68-69: This is suggested in Fettweis et al. (2013), not Hofer et al. (2017).

- P5L133-134: Performance of models does not necessarily improves with sophistication. Has this been evaluated? For which specific processes? Is there a reference to support this assertion? If not, I'd recommend to remove it.

- P5L141: The remaining bias in the downscaled products is of much greater interest (and actually required) for your study than the bias in the original forcing fields.

- P5L142: Prefer "negative" over "cold" bias. The model is too cold when the temperatures are too low. Idem for warm bias.

- P5L162: RMSE is a valuable complement to the other statistical indexes used here as it enables quantification of the model performance relative to each individual values of

the dataset. I'd advise to align with the description made for Greenland and give statistics for the Antarctic SMB too.

- P6L164-165: At the resolution used over Antarctica here (0.11°), Mottram et al. (2020) report a mean bias of -2.1°C.

- P7L198 : Why is a "simplified equation for surface mass balance" used here ? Not sure what you mean.

- P7L209-210: Why not using a mask? Ice shelves are peripheral regions of the ice sheet, likely experiencing strong melt rates compared to inner regions. You could give a stronger, more accurate meaning to this result by removing them from your calculation.

- P7L214: Why don't you simply speak of SMB here? Homogenizing the definition of SMB within the whole manuscript would help improve readability.

- P8L223: That is, much more runoff than the intermodel estimates in Mottram et al; (2020) over the present climate, and even more than what is projected by some GCM-driven RCM estimates for the end of the century (see Kittel et al. 2020). Please discuss these very unusual (and most likely, unlikely) results.

- P8L230: Only 1°C higher could not solely explain such large increases in runoff.

- P9L253-255: in Mottram et al. (2020), HIRHAM5 model has been used for downscaling ERA-I, not EC-Earth. Generally, prefer the word evaluated over validated as significant model bias remain, and comment on the positive aspects of the evaluation. Note that the performance of an RCM is also relative to the forcing, and does not guarantee a similar performance with different forcings, which explains the very different results presented here when compared to Mottram et al. (2020). This is notably why an evaluation per forcing is required.

- P9L258: These two studies focuses on Antarctica, so how they could relate to the results presented here for Greenland is confusing.

- P9L258-260: Give the corresponding references of Fettweis et al. (2013), Hofer et al. (2020) and Kittel et al. (2020) to improve clarity. Prefer naming the models used by the authors rather than the authors if you refer to the models.

- P9L262-264: This statement requires a more detailed argumentation, or ideally an analysis of the model outputs to help making the connection with Holland et al. (2019) (which is missing in the references).

- P10L284-286: You could and should use a mask, to remove the SMB over ice shelves which does not contribute to SLR, before converting to SLR equivalent.

- Table 2: Specify if these numbers relate to total or grounded ice sheet area.

**References**

Agosta, C., Favier, V., Krinner, G., Gallée, H., Fettweis, X. and Genthon, C.: High-315 resolution modelling of the Antarctic surface mass balance, application for the twentieth, twenty first and twenty second centuries, Climate Dynamics, 41(11–12), 3247–3260, doi:10.1007/s00382-013-1903-9, 2013.

Agosta, C., Amory, C., Kittel, C., Orsi, A., Favier, V., Gallée, H., Van den Broeke, M., Lenaerts, J., Wessem, J. M., Berg, W. J. and Fettweis, X.: Estimation of the Antarctic surface mass balance using the regional climate model MAR (1979–2015) and identification of dominant processes, The Cryosphere, 13, 281–296, doi:10.5194/tc-13-281-2019, 2019.

Delhasse, A., Fettweis, X., Kittel, C., Amory, C., and Agosta, C.: Brief communication: Impact of the recent atmospheric circulation change in summer on the future surface mass balance of the Greenland Ice Sheet, The Cryosphere, 12, 3409–3418, https://doi.org/10.5194/tc-12-3409-2018, 2018.

Fettweis, X., Franco, B., Tedesco, M., van Angelen, J. H., Lenaerts, J. T. M., van den Broeke, M. R., and Gallée, H.: Estimating the Greenland ice sheet surface mass balance contribution to future sea level rise using the regional atmospheric climate model MAR, The Cryosphere, 7, 469–489, https://doi.org/10.5194/tc-7-469-2013, 2013.

Hofer, S., Tedstone, A. J., Fettweis, X., and Bamber, J. L.: Decreasing cloud cover drives the recent mass loss on the Greenland Ice Sheet, Science Advances 3(6), e1700584, DOI: 10.1126/sciadv.1700584, 2017.

Hofer, S., Lang, C., Amory, C., Kittel, C., Delhasse, A., Tedstone, A., and Fettweis, X.: Greater Greenland Ice Sheet contribution to globalsea level rise in CMIP6, Nat. Com., 9, 523–528, 2020, https://www.nature.com/articles/s41467-020-20011-8

Holland, P.R., Bracegirdle, T.J., Dutrieux, P. et al. West Antarctic ice loss influenced by internal climate variability and anthropogenic forcing. Nat. Geosci. **12,** 718–724 (2019). https://doi.org/10.1038/s41561-019-0420-9.

Kittel, C., Amory, C., Agosta, C., Jourdain, N. C., Hofer, S., Delhasse, A., Doutreloup, S., Huot, P.-V., Lang, C., Fichefet, T., and Fettweis, X.: Diverging future surface mass balance between the Antarctic ice shelves and grounded ice sheet, The Cryosphere Discuss. [preprint], https://doi.org/10.5194/tc-2020-291, in review, 2020.

Ligtenberg, S., Berg, W. J., Van den Broeke, M., Rae, J., and Meijgaard, E.: Future surface mass 410 balance of the Antarctic ice sheet and its influence on sea level change, simulated by a regional atmospheric climate model. Climate Dynamics. 41. 10.1007/s00382-013-1749-1, 2013.

Mottram, R., Hansen, N., Kittel, C., van Wessem, M., Agosta, C., Amory, C., Boberg, F., van de Berg, W. J., Fettweis, X., Gossart, A., van Lipzig, N. P. M., van Meijgaard, E., Orr, A., Phillips, T., Webster, S., Simonsen, S. B., and Souverijns, N.: What is the Surface Mass Balance of Antarctica? An Intercomparison of Regional Climate Model Estimates, The Cryosphere Discuss., https://doi.org/10.5194/tc-2019-333, in review, 2020.

Noël, B., van Kampenhout, L., Lenaerts, J.T.M., van de Berg, W. J., and van den Broeke, M. R.: A 21$^{st}$ Century Warming Threshold for Sustained Greenland Ice Sheet Mass Loss, Geophys. Res. Let., in press, 2021.

Sherwood, S., Webb, M., Annan, J., Armour, K., Forster, P., Harg-reaves, J., Hegerl, G., Klein, S., Marvel, K., Rohling, E., Watan-abe, M., Andrews, T., Braconnot, P., Bretherton, C. S., Foster, G.L., Hausfather, Z., von der Heydt, A. S., Knutti, R., Mauritsen, T.,Norris, J. R., Proistosescu, C., Rugenstein, M., Schmidt, G. A.,Tokarska, K. B., and Zelinka, M. D.: An assessment of Earth'sclimate sensitivity using multiple lines of evidence, Rev. Geo-phys., https://doi.org/10.1029/2019RG000678, 2020.

Zhu, J., Poulsen, C. J., & Otto, Bliesner, B. L. (2020). High climate sensitivity in CMIP6 model not supported by paleoclimate. Nature Climate Change, 10(5), 378–379. https://doi.org/10.1038/s41558-020-0764-6.